# Selective recruitment of the cerebellum evidenced by task-dependent gating of inputs

Ladan Shahshahani[1,2]*, Maedbh King[3], Caroline Nettekoven[1], Richard B Ivry[4,5], Jörn Diedrichsen[1,6,7]*

[1]Western Institute for Neuroscience, Western University, London, Ontario, Canada; [2]Cognitive, Linguistics, & Psychological Science, Brown University, Providence, United States; [3]McGovern Institute, Massachusetts Institute of Technology, Cambridge, United Kingdom; [4]Department of Psychology, University of California, Berkeley, Berkeley, United States; [5]Helen Wills Neuroscience Institute, University of California Berkeley, Berkeley, United States; [6]Department of Statistical and Actuarial Sciences, Western University London, Ontario, Canada; [7]Department of Computer Science, Western University, London, Ontario, Canada

**\*For correspondence:**
ladan_shahshahani@brown.edu (LS);
jdiedric@uwo.ca (JD)

**Abstract** Functional magnetic resonance imaging (fMRI) studies have documented cerebellar activity across a wide array of tasks. However, the functional contribution of the cerebellum within these task domains remains unclear because cerebellar activity is often studied in isolation. This is problematic, as cerebellar fMRI activity may simply reflect the transmission of neocortical activity through fixed connections. Here, we present a new approach that addresses this problem. Rather than focus on task-dependent activity changes in the cerebellum alone, we ask if neocortical inputs to the cerebellum are gated in a task-dependent manner. We hypothesize that input is upregulated when the cerebellum functionally contributes to a task. We first validated this approach using a finger movement task, where the integrity of the cerebellum has been shown to be essential for the coordination of rapid alternating movements but not for force generation. While both neocortical and cerebellar activity increased with increasing speed and force, the speed-related changes in the cerebellum were larger than predicted by an optimized cortico-cerebellar connectivity model. We then applied the same approach in a cognitive domain, assessing how the cerebellum supports working memory. Enhanced gating was associated with the encoding of items in working memory, but not with the manipulation or retrieval of the items. Focusing on task-dependent gating of neocortical inputs to the cerebellum offers a promising approach for using fMRI to understand the specific contributions of the cerebellum to cognitive function.

## eLife assessment

This **important** study reports a novel approach to studying cerebellar function based on the idea of selective recruitment using fMRI. It provides **convincing** evidence for task-dependent gating of neocortical input to the cerebellum during a motor task and a working memory task. The study will be of interest to a broad cognitive neuroscience audience.

## Introduction

More than 30 years of neuroimaging has revealed that the human cerebellum is activated in a broad range of tasks including motor (*Spraker et al., 2012*), language (*Petersen et al., 1989*), working

memory (*Marvel and Desmond, 2010*), attention (*Allen et al., 1997*), social (*Van Overwalle et al., 2015*), and visual cognition tasks (*van Es et al., 2019*) – for a review see *Diedrichsen et al., 2019*. Indeed, there are very few tasks that do not lead to activity in some part of the cerebellum. The presence of cerebellar activity is usually taken as evidence that the cerebellum plays a functional role associated with these tasks.

However, there is an important problem with this line of reasoning. The cerebellar blood-oxygen-level-dependent (BOLD) signal does not reflect the activity levels of Purkinje cells, the output of the cerebellar cortex (*Caesar et al., 2003a*; *Thomsen et al., 2004*; *Thomsen et al., 2009*). Rather, it is determined solely by mossy fiber (*Akgören et al., 1994*; *Gagliano et al., 2022*; *Mapelli et al., 2017*) and climbing fiber (*Caesar et al., 2003b*; *Mathiesen et al., 2000*) input, with the former likely playing the dominant role (*Attwell and Iadecola, 2002*; *Howarth et al., 2012*).

Mossy fibers carry input from a wide array of neocortical areas, including prefrontal and parietal association cortices, as demonstrated directly through viral tracing studies in non-human primates (*Kelly and Strick, 2003*), and indirectly through resting-state functional connectivity (rs-FC) analysis in humans (*Buckner et al., 2011*; *Ji et al., 2019*; *Marek et al., 2018*; *O'Reilly et al., 2010*). This means that increases in the cerebellar BOLD signal could simply reflect the automatic transmission of neocortical activity through fixed anatomical connections. As such, whenever a task activates a neocortical region, the corresponding cerebellar region would also be activated, regardless of whether the cerebellum is directly involved in the task or not.

The preceding arguments suggest that it is important to consider cerebellar activation in the context of the neocortical regions that provide its input. To approach this problem, we have recently developed and tested a range of cortical–cerebellar connectivity models (*King et al., 2023*), designed to capture fixed, or task-invariant, transmission between neocortex and cerebellum. For each cerebellar voxel, we estimated a regularized multiple regression model to predict its activity level across a range of task conditions (*King et al., 2019*) from the activity pattern observed in the neocortex for the same conditions. The models were then evaluated in their ability to predict cerebellar activity in novel tasks, again based only on the corresponding neocortical activity pattern. Two key results emerged from this work. First, while rs-FC studies (*Buckner et al., 2011*; *Ji et al., 2019*; *Marek et al., 2018*) have assumed a 1:1 mapping between neocortical and cerebellar networks, models which allowed for convergent input from multiple neocortical regions to a single cerebellar region performed better in predicting cerebellar activity patterns. Second, when given a cortical activation pattern, the best performing model could predict about 50% of the reliable variance in the cerebellar cortex across tasks (*King et al., 2023*).

This model offers a powerful null model to evaluate whether the cerebellar BOLD signal can be fully explained by the fixed transmission of input from neocortex in a task-invariant manner. The fact that the prediction of these models did not reach the theoretically possible prediction accuracy suggests that the connectivity between the neocortex and cerebellum may not be fully task-invariant. Instead, neocortical input to the cerebellum may be modulated as a function of the relative importance of cerebellar computation in a task-specific manner. We refer to this as the *selective recruitment* hypothesis; specifically, we hypothesize that input is upregulated when cerebellar computation is required. Such task- or state-dependent gating would make evolutionary sense, given the substantial metabolic cost of granule cell activity (*Attwell and Iadecola, 2002*; *Howarth et al., 2010*).

To evaluate the selective recruitment hypothesis, we first turned to the motor domain where clinical studies provide a strong a priori hypothesis of when the cerebellum should be selectively recruited. Patients with cerebellar damage consistently show impairments in performing rapid alternating movements, a symptom called dysdiadochokinesia (*Hallett et al., 1991*; *Mai et al., 1988*). In contrast, these patients are generally able to exert grip forces comparable to healthy controls (*Mai et al., 1988*). Based on the selective recruitment hypothesis, we predicted that increases in cerebellar BOLD will be greater for increases in tapping speed compared to increases in finger force output, even when the neocortical activity is matched between conditions.

We then applied the approach in a cognitive domain. Working memory tasks have been shown to robustly activate hemispheric regions of cerebellar lobules VI, VII, and VIII (*Chen and Desmond, 2005*; *Desmond et al., 1997*). Furthermore, patients with cerebellar damage tend to show deficits in verbal working memory tasks (*Cooper et al., 2012*; *Ilg et al., 2013*; *Kansal et al., 2017*; *Peterburs et al., 2010*; *Ravizza et al., 2006*). However, the form of the deficits is unclear and quite variable (*Hokkanen*

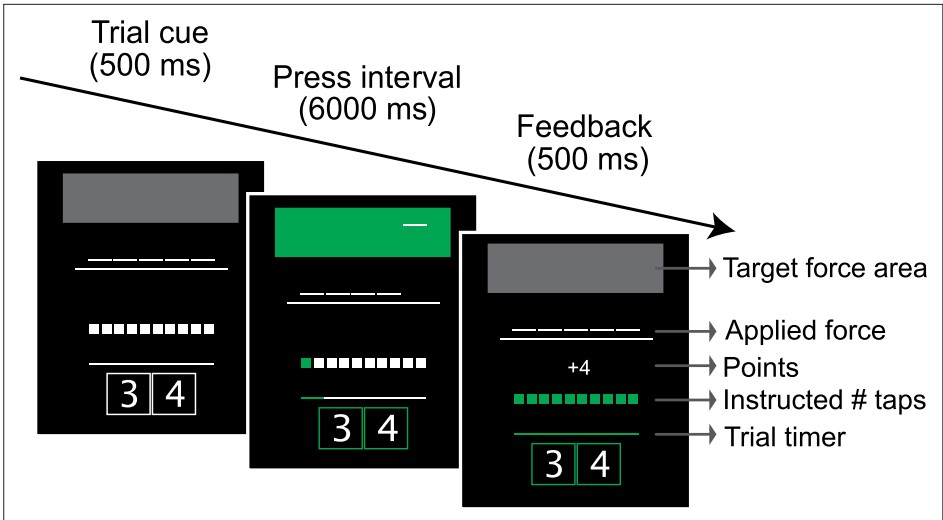

**Figure 1.** Timeline of events in the alternating finger tapping task. The height of the target force area indicated the target force, the number of white squares the target number of taps. During the press interval, the participant alternatively tapped the middle and ring finger. After each tap, the next box turned green. Reward feedback (e.g.,+4) was based on their performance.

et al., 2006; **McDougle et al., 2022**; **Pleger and Timmann, 2018**; **Starowicz-Filip et al., 2021**), making it difficult to draw inferences concerning the computational contribution of the cerebellum to working memory tasks. We designed a digit span task that allowed us to evaluate three factors relevant for working memory: (1) task phase (encoding/recall), (2) memory load, and (3) information manipulation (forward/backward recall). We asked which combination of these three factors leads to selective recruitment in cerebellar working memory regions.

## Results
### Motor task

To test the selective recruitment hypothesis in the motor domain, we used a task which involved alternating finger presses of middle and ring finger (**Figure 1**). Starting at a baseline level of 1 Hz and 2.5 N, we either increased the force of each response or the required rate (**Table 1**). Both manipulations are expected to produce an increase in the BOLD response in neocortical motor areas (**Diedrichsen et al., 2013**; **Thickbroom et al., 1998**). As such, our task-invariant connectivity model predicts increased cerebellar activity with both increases in speed and force (**Spraker et al., 2012**). Critically, selective recruitment predicts that for equivalent activity levels in the neocortex, cerebellar activity should be higher in the speed than in the force condition.

Participants complied well with task instructions, as evidenced by the group-averaged peak forces and number of taps, which were close to the target values (**Table 1**). The high error rate for the

**Table 1.** Mean and between-subject standard deviation (±) of force, speed, and error rate for each condition across subjects.

| Condition | Target force (N) | Target # taps in 6 s | Average force (N) | Average # taps in 6 s | Error rate (%) |
|---|---|---|---|---|---|
| High speed | 2.5 | 18 | 2.93 ± 0.48 | 17.72 ± 0.84 | 5 ± 0.21 |
| Medium speed | 2.5 | 10 | 2.84 ± 0.45 | 10.12 ± 0.44 | 1 ± 0.12 |
| Baseline | 2.5 | 6 | 2.80 ± 0.41 | 6.32 ± 0.8 | 15 ± 0.36 |
| Medium force | 6 | 6 | 6.10 ± 0.49 | 6.04 ± 0.2 | 4 ± 0.18 |
| High force | 10 | 6 | 9.73 ± 0.66 | 6.04 ± 0.2 | 2 ± 0.13 |

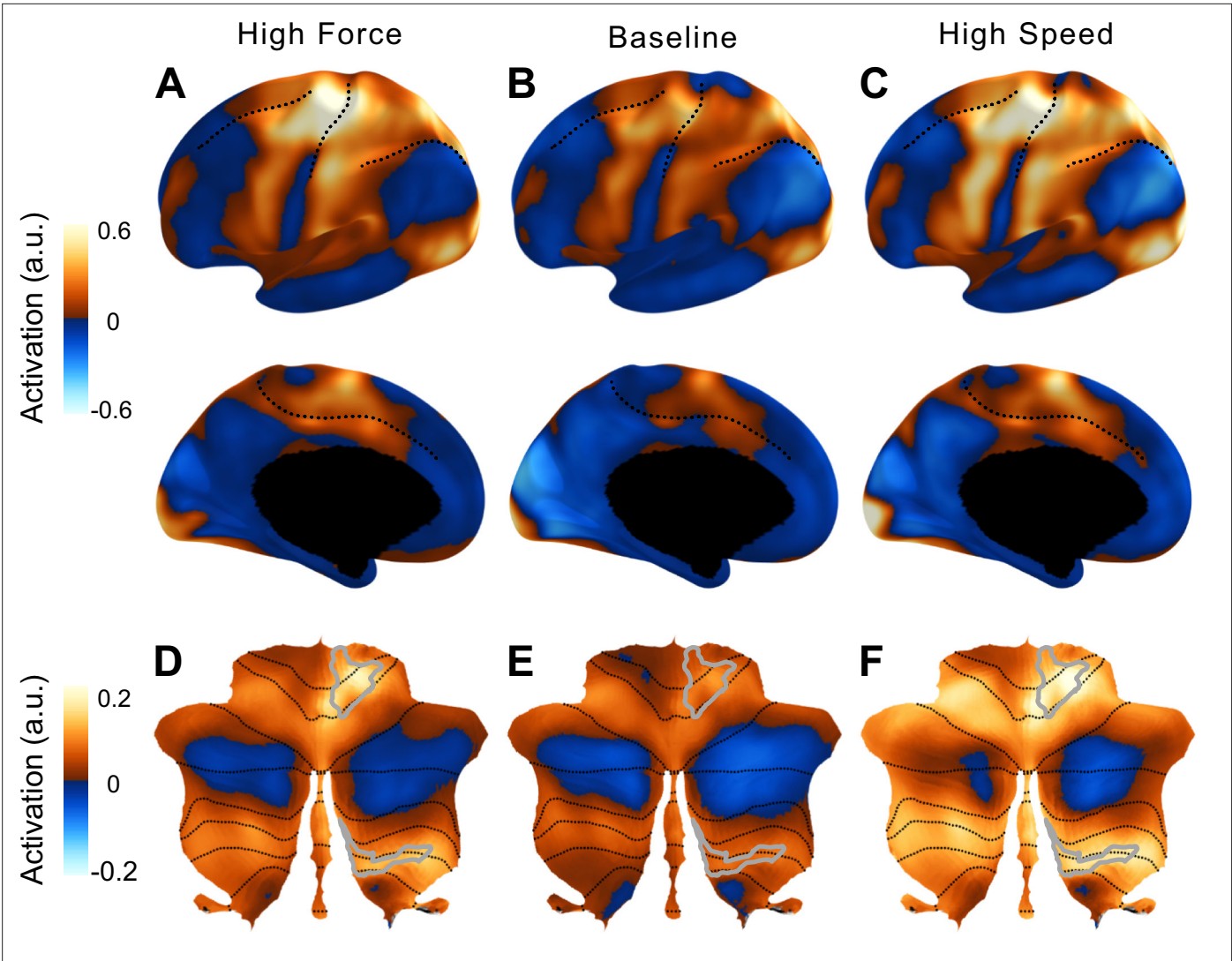

**Figure 2.** Activation in the cortico-cerebellar motor network compared to rest. Activity maps for high-force (left), baseline (middle), and high-speed (right) conditions. High levels of force and speed were chosen to show the spatial distribution of activity. Medium level of force and speed resulted in similar maps with activity levels between the baseline and high conditions. (**A–C**) Lateral and medial surface of the left hemisphere. Dotted lines indicate the superior frontal, central, intra-parietal, and cingulate sulcus. (**D–F**) Flat map of the cerebellum (*Diedrichsen and Zotow, 2015*) with lobular boundaries indicated in dotted line. The right anterior and posterior hand motor area (M3R, gray outline) was defined by a new functional atlas of the cerebellum (*Nettekoven et al., 2024b*).

baseline condition reflects the fact that some of the participants completed the six taps in less than the minimum interval of 4 s in this very easy condition.

### Increasing force and speed leads to increased activation in cortico-cerebellar motor network

As expected for right-hand movements, activation was observed in the hand areas of left (contra-lateral) M1 and S1 (*Figure 2*). Compared to the baseline condition, the combined M1/S1 region of interest (ROI) showed a significant activation increase in the high-force ($t_{15}$ = 9.41, p = 1.10 × 10$^{-7}$) and the high-speed conditions ($t_{15}$ = 8.29, p = 5.54 × 10$^{-7}$). Similarly, activity in the right anterior and posterior motor areas of the cerebellum (outlined in light gray in *Figure 2*, see *Methods* for details on ROI) increased with increasing force ($t_{15}$ = 14.21, p = 4.14 × 10$^{-10}$) and speed ($t_{15}$ = 7.60, p = 1.59 × 10$^{-6}$). The medium force and speed conditions were between baseline and high conditions, replicating

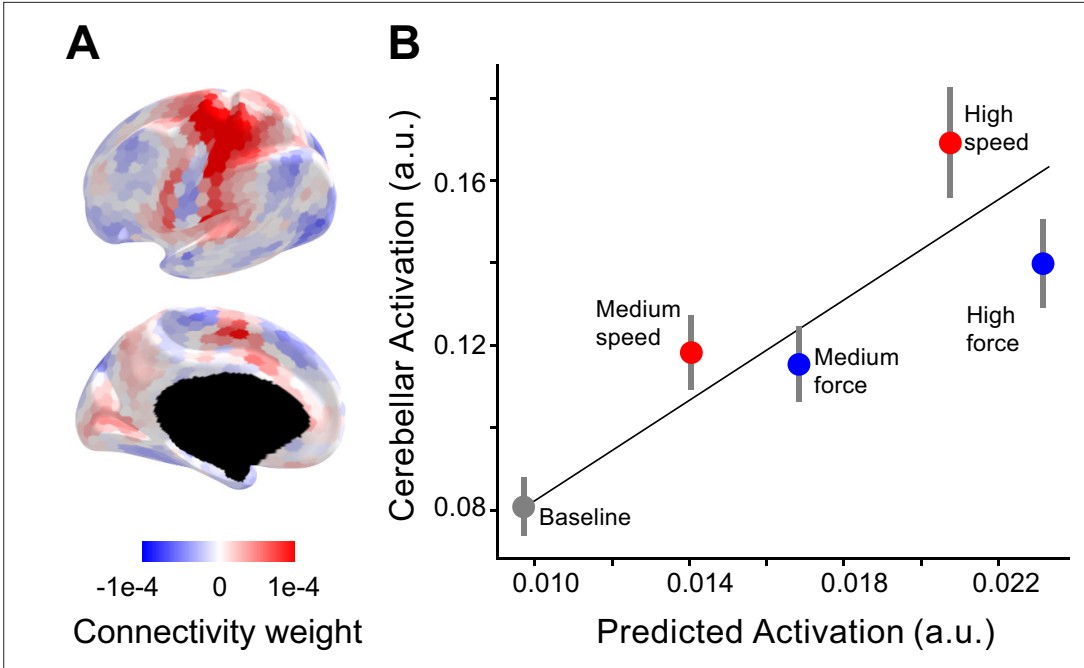

**Figure 3.** Selective recruitment of cerebellum for fast alternating finger movements. (**A**) Average connectivity weights from a group-level connectivity model (Ridge regression, multi-domain task battery [MDTB], task set A) for the cerebellar right-hand area shown on inflated surface of the left hemisphere. For evaluation of alternative connectivity models see *Figure 3—figure supplement 1*. (**B**) Average observed cerebellar activation (*y*-axis) plotted against average prediction from the connectivity model (*x*-axis). Resting baseline (located at 0,0) is not shown explicitly but included in the regression. The error bars indicate the standard error of the mean of the signed residuals.

The online version of this article includes the following figure supplement(s) for figure 3:

**Figure supplement 1.** Connectivity models evaluation.

previous findings of a parametric modulation of activity with both force (*Spraker et al., 2012*) and speed (*Jäncke et al., 1999*).

Visual inspection of the activation maps (*Figure 2D vs. 2F*) suggests that cerebellar activity increased more with speed than with force. One might take this result alone as an indication that recruitment of the cerebellum is relatively greater when the task requires the coordination of rapid finger movements compared to when an increase in force is required. However, the neocortical activation patterns for speed and force conditions were not completely matched (*Figure 2A vs. 2C*): Increasing speed led to more widespread activation in secondary motor areas compared to increasing force. Therefore, the observed differences in cerebellar activity could have resulted from additional fixed inputs from premotor and supplementary motor areas, rather than from a task-dependent recruitment of cerebellar circuits for the speed task.

## Cerebellar activity for increased speed is larger than predicted by task-invariant connectivity

To distinguish these two hypotheses, we used our task-invariant cortico-cerebellar connectivity model (L2-regularized multiple regression, see methods), trained on a separate set of participants across a large range of tasks (*King et al., 2023*). This model provides an estimate of cerebellar activity expected from fixed anatomical connections with the neocortex. We take this as the reference point for asking if observed activation levels are greater than expected; what we use as our operational definition of selective recruitment. *Figure 3A* shows the connectivity weights from this model for the cerebellar right-hand area, region M3 (*Nettekoven et al., 2024b*). According to the model, inputs to cerebellar M3 do not only come from contralateral M1 and S1, but also from premotor and supplementary motor regions.

We multiplied the neocortical activity patterns from each individual and condition with the connectivity weights from the model to predict the corresponding cerebellar M3 activity level. Note that the connectivity weights were estimated on subjects from independent task-based functional magnetic resonance imaging (fMRI) datasets; therefore, the predicted values were on a different scale compared to the observed values (*Figure 3B*). To account for this scaling difference, we used a simple linear regression between observed and predicted values.

In general, the predicted values closely match the observed values (average $R^2$ = 0.60, standard error of the mean = 0.01). However, relative to the force conditions, the speed conditions resulted in larger cerebellar activity, even though the predicted activity was smaller. To test for systematic deviations across subjects, we submitted the signed residuals for all conditions to a one-way analysis of variance (ANOVA), revealing a significant effect of condition ($F_{4, 60}$ = 6.796, p = 1.1 × 10$^{-4}$). Post hoc tests revealed that the signed residual for the high-speed condition was significantly higher than for the high-force condition ($t_{15}$ = 2.37, p = 0.0157). This was also the case when comparing medium speed and medium force ($t_{15}$ = 1.94, p = 0.035). In summary, the increases in cerebellar activity for speed outstripped the activity increases for force, even when we accounted for differences in activity for the two conditions in neocortical input regions.

## Alternative connectivity models

We recognize that our results depend on the connectivity model used to predict cerebellar activity. To ensure that our findings were robust, we replicated the results using two additional connectivity models (*Figure 3—figure supplement 1*). First, we used an L1-regularized model, which resulted in sparser connectivity weights. In our previous study, we found that this model performed only slightly worse in predicting left-out data compared to the L2-regularized model (*King et al., 2023*). Second, we used a connectivity model that was trained on the entire multi-domain task battery (MDTB) dataset plus four additional large task-based datasets (Fusion model, see Methods). For both connectivity models, the predicted difference in the residual for the high-speed vs. high-force condition remained significant (L1 regression: $t_{15}$ = 2.373, p = 0.0315, Fusion model: $t_{15}$ = 2.140, p = 0.0492). Thus, consistent across various connectivity models, the results indicate selective recruitment of the cerebellum when the demands on finger coordination are increased relative to when the demands on force output are increased.

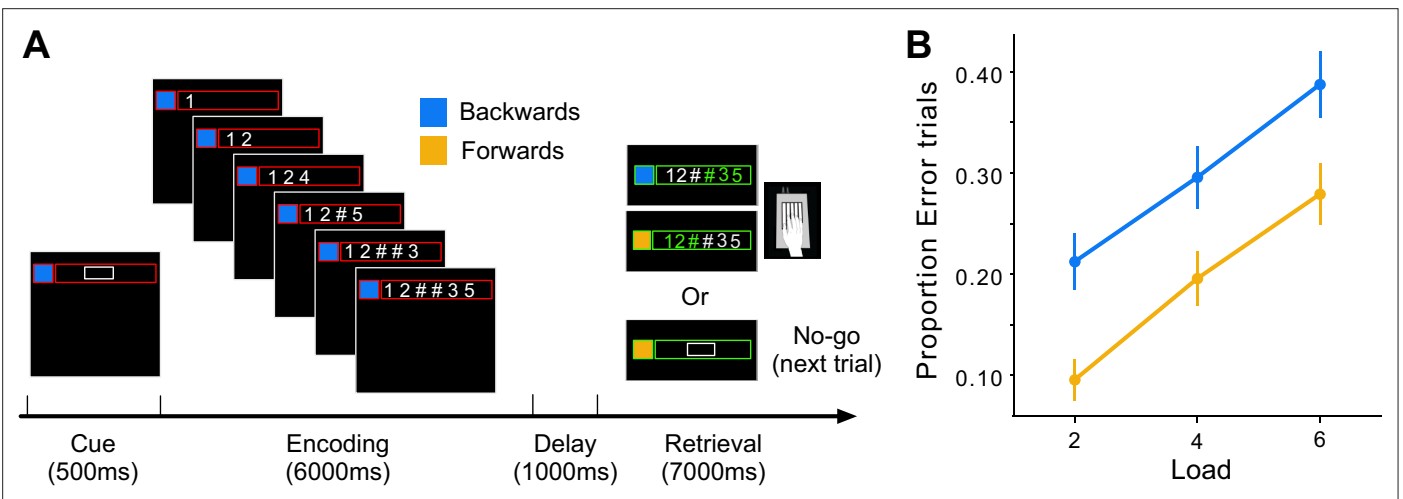

**Figure 4.** The digit span task and behavioral performance. (**A**) Timeline of trial events. The cue signaled the recall direction (blue for backward and yellow for forward) and memory load (size of the white box indicated the number of memory digits) of the upcoming trial. During encoding, a new digit appeared every second and was replaced by the # symbol if it was a memory digit. After a 1-s delay, the task progressed to either the retrieval phase (Go trial) or skipped directly to the next trial (No-Go trials). (**B**) Proportion of error trials. Error bars indicate standard error of the mean across participants.

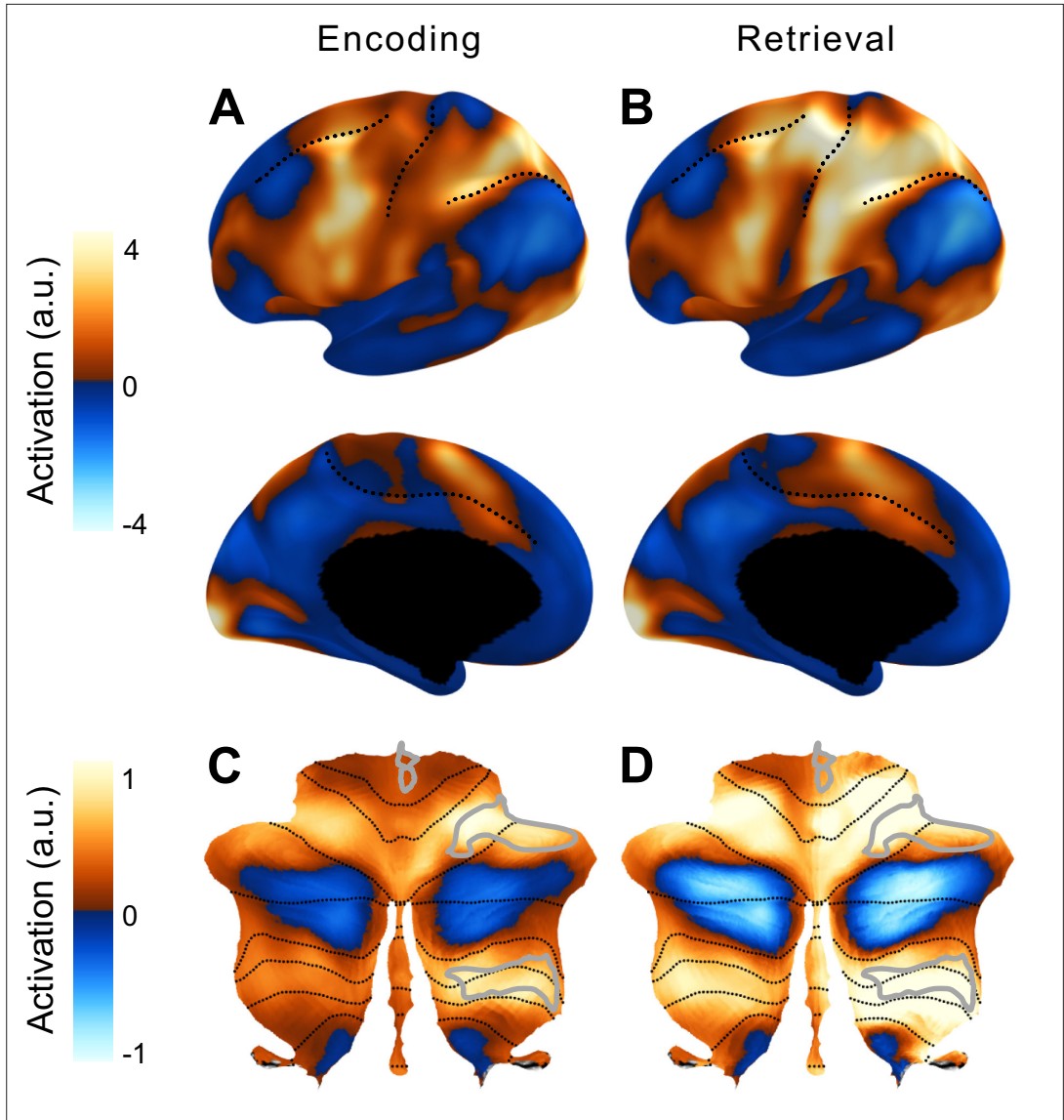

**Figure 5.** Average activation in the cortico-cerebellar network for working memory. Group-averaged activation during the encoding (**A**) and retrieval (**B**) phases on an inflated representation of the left cerebral hemisphere (as in *Figure 2*). (**C**, **D**) Group average activity during the two phases in the cerebellum. The D3R subregion of the multi-demand network in the right cerebellar hemisphere was used in the main analysis (outlined in light gray).

## Working memory task

We conducted our initial test of the selective recruitment hypothesis using a motor task, for which we had a strong a priori prediction concerning the factors that lead to an upregulation of cortical input to the cerebellum. Having validated our approach here, we next turned to the cognitive domain, asking if our approach could help shed light on the functional contribution of the cerebellum to verbal working memory. Cerebellar activation, particularly in Lobules VI, Crus I, and VIII is consistently observed in fMRI studies of working memory (*Cohen et al., 1997*; *Courtney et al., 1997*; *D'Esposito and Postle, 2015*; *Nee et al., 2013*). Here, we test if these cerebellar areas are especially recruited for a specific component process of working memory.

We implemented a digit span task in which participants memorized and subsequently recalled a sequence of visually presented digits (*Figure 4A*). Each trial began with a cue that signaled the recall direction (forward or backward) and the number of digits that had to be remembered (2, 4, 6). During the encoding phase, six digits were sequentially displayed from left to right at a rate of 1 digit/s. At

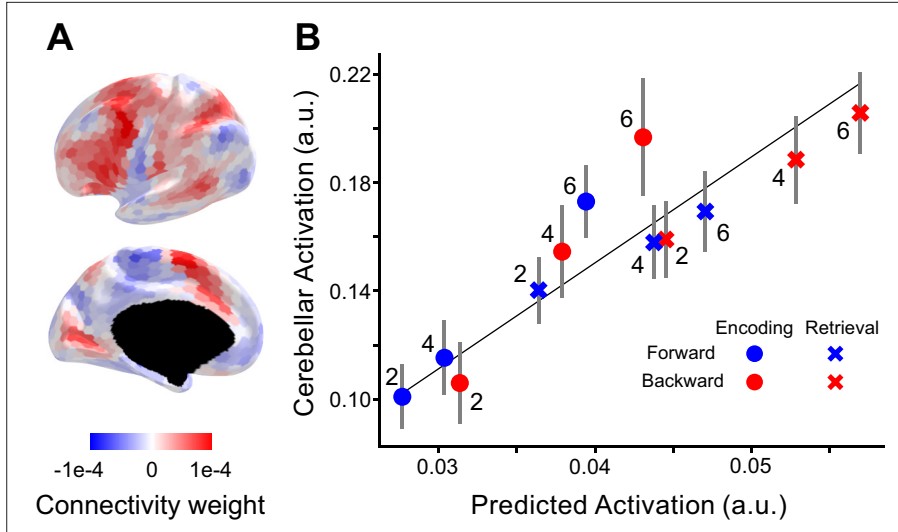

**Figure 6.** Selective recruitment of cerebellum in digit span task. (**A**) Average connectivity weights from a group-level connectivity model for the cerebellar D3R region of interest. (**B**) Average observed cerebellar activation (*y*-axis) plotted against average prediction from the connectivity model (*x*-axis). Line shows the best linear relationship between predicted and observed activity with an intercept of zero. Error bars show standard error of the mean (SEM) of the signed residuals for each condition across subjects.

the end of each 1-s presentation interval, the next digit was presented and the most recent digit either remained on the screen or was replaced by the hashtag symbol (#) if it had to be remembered. During the retrieval phase, all six digits had to be typed in on the keyboard, either backwards or forwards. Thus, while the memory load varied between two and six items, all conditions involved the presentation and production of six digits. On 25% of the trials (No-Go) the trial was terminated at the start of retrieval phase. The inclusion of these encoding-only trials gave us separate estimates of activity related to the encoding and retrieval phases (see methods). In summary, we measured activity for 12 conditions (2 recall directions × 3 memory loads × 2 phases).

*Figure 4B* shows the error rate (trials with at least one wrong press) during the scanning session. As expected, error rates increased with memory load and were also higher in the backwards condition. Consistent with previous imaging studies, the verbal working memory task led to high activity in the fronto-parietal network (*Cohen et al., 1997*; *Courtney et al., 1997*; *D'Esposito and Postle, 2015*; *Nee et al., 2013*; *Owen et al., 2005*) during the encoding and retrieval phases (*Figure 5*). During the latter phase, we also observed activation in cortical motor areas, reflecting the response requirements of the task. Within the cerebellum, encoding and retrieval activated a superior region (lateral parts of lobule VI, extending to Crus I), as well as an inferior region (VIIb and VIIIa) (*Chen and Desmond, 2005*; *Desmond et al., 1997*). As observed previously for verbal working memory, the activity was more pronounced in the right than in the left cerebellar hemisphere (*Desmond and Fiez, 1998*). In our symmetric functional atlas (*Nettekoven et al., 2024b*), the best corresponding functional region was right D3 (see *Methods, ROI*).

## Cerebellar activity for encoding at high load is larger than predicted by task-invariant connectivity

To estimate which neocortical regions provide input to our cerebellar ROI (D3R), we again used the task-invariant model of cortico-cerebellar connectivity model (*King et al., 2023*). The connectivity weights from the model (*Figure 6A*) suggest converging input from area 55b (located at the inferior end of the middle frontal gyrus), premotor eye field, area 6r (anterior to the primary motor cortex), and supplementary and cingulate eye field (SCEF, dorsomedial frontal cortex; *Glasser et al., 2016*). The model was used to predict activity in the cerebellar ROI for each condition and participant. After fitting a linear regression to account for scale differences between predicted and observed activations, we found that the predicted values matched the observed values relatively well at the individual level ($R^2 = 0.42$, standard error = 0.01).

Turning to our test of selective recruitment, there was one clear deviation where the observed cerebellar activation was greater than predicted (*Figure 6B*): During encoding in the highest load condition. A repeated 1-factor measures ANOVA on the residuals across all 12 conditions found a systematic deviation across participants ($F_{11, 165}$ = 2.22, p = 0.0156). When we analyze the residuals using a 2 (phase) × 2 (recall direction) × 3 (load) ANOVA, we found a significant two-way interaction between load and phase ($F_{2, 30}$ = 4.38, p = 0.02), but no significant effect of recall direction ($F_{1, 15}$ = 0.95, p = 0.34). Thus, the results indicate selective recruitment of the cerebellum when the number of items to be encoded into working memory is high, an effect that holds for both the forward and backwards conditions.

As with the motor task, we repeated the analyses using two other cortico-cerebellar connectivity models. The same pattern of results was found with a significant difference across conditions (L1-regularized model trained on the MDTB dataset: $F_{11, 165}$ = 2.34, p = 0.0105; L2-regularized model trained on five datasets: $F_{11, 165}$ = 2.55, p = 5.3 × 10$^{-3}$) due to a significant deviation from the predicted level of activation during encoding with a load of 6.

## Discussion

Functional neuroimaging studies have shown that the human cerebellum is activated across a broad range of task domains. However, before inferring that the cerebellum contributes causally to these task, we need to consider that the BOLD signal in the cerebellar cortex is likely dominated by mossy-fiber input (*Alahmadi et al., 2015*; *Alahmadi et al., 2016*; *Gagliano et al., 2022*; *Mapelli et al., 2017*; *Mathiesen et al., 2000*; *Thomsen et al., 2004*; *Thomsen et al., 2009*), which in humans mostly carry information from neocortex. Thus, BOLD activation changes observed in the cerebellar cortex could be a consequence of the transmission of information through fixed anatomical connections. Given this possibility, it is problematic to assume that the activation implies a significant functional contribution, let alone make inferences about the nature of that contribution.

Our first experiment in the motor domain clearly illustrates this problem. We found highly significant increases in the cerebellar BOLD signal for increases in both movement speed and force. Using the inferential logic traditionally employed in neuroimaging, one might conclude that the cerebellum plays a functional role in regulating both parameters. However, clinical studies have shown that cerebellar pathology results in a marked impairment in the ability to produce fast alternating movements, but has little impact on maximal force generation (*Mai et al., 1988*). This suggests that some observed change in cerebellar activity may not be functionally involved in controlling behavior.

To address these concerns, we first needed a strong null model: We used a cortico-cerebellar connectivity model (*King et al., 2023*), which was optimized to predict cerebellar activity based only on neocortical activity patterns across a wide array of tasks; thus, it provides a prediction of the expected cerebellar activity for any task if all cerebellar activity was indeed caused by a fixed, task-invariant transmission of activity from neocortex. This prediction takes into account that some functional networks, such as the fronto-parietal and salience networks, occupy a relatively larger area of the cerebellum than of the neocortex (*Buckner et al., 2011*; *Marek et al., 2018*), and that there will be variation in convergence across the cerebellar cortex. As shown in our previous study (*King et al., 2023*), this model provides a good prediction of cerebellar activity across a broad range of tasks, including those not used in developing the model. This confirms that a large proportion of the observed variation of cerebellar activity across tasks can be accounted for by fixed functional connections between neocortical and cerebellar regions.

The central idea explored in the current paper is that systematic deviations from this null model would occur, if neocortical input was upregulated when the cerebellum is required for a task (and/or downregulated when it is not). We first tested this selective recruitment hypothesis in the motor domain, where we had the strong a priori prediction: If input to the cerebellum is gated in a task-specific manner, then it should be upregulated during the production of fast alternating finger movements as compared to the production of high forces with the same fingers. This was indeed the case; activity in the cerebellar hand area increased more for increasing speed than force, even when activity in the neocortical hand areas was approximately matched across conditions. These results provide clear evidence for task-dependent gating (*Cole et al., 2021*).

This phenomenon now offers a new, more stringent criterion to infer functional involvement of the cerebellum: Rather than focusing on activation for a given task per se, we can now test if a cerebellar

area is selectively recruited for a task. If the input to the cerebellum is upregulated in a task-specific manner, then the observed cerebellar activity should be larger than precited using the simultaneously observed neocortical activity and a task-invariant connectivity model. We can apply this approach to cognitive and social task to provide new insights into the contribution of the cerebellum in these domains.

As a first application, we chose to investigate cerebellar activation during a working memory task. Previous studies have consistently shown deficits in verbal working memory in cerebellar patients (*Ilg et al., 2013*; *Kansal et al., 2017*; *Ravizza et al., 2006*). The exact nature of these deficits, however, is still a matter of considerable debate. Here we tested whether a variation in memory load, recall direction (requiring reversal of digits during backward recall), task phase (encoding vs. retrieval), or some combination of these factors, would lead to selective recruitment of the cerebellum. We found strong activity in cerebellar working memory regions across all conditions, with the retrieval phase for six items recalled in reversed order leading to most activation. Following the traditional neuroimaging inference approach, these results would be taken as evidence for cerebellar involvement in item manipulation during retrieval.

Our new analysis, however, highlights that the corresponding neocortical working memory regions also showed the highest activation level during this condition and, importantly, that the cerebellar activity in this condition was well predicted using a task-invariant connectivity model. In contrast, the connectivity-based analysis identified the encoding of six items into working memory (for both forward and backward retrieval) as a condition for which the observed cerebellar activity outstripped the prediction by the null model. This suggests that the cerebellum has a special role in encoding larger item sets into working memory. Further experiments using other working memory tasks and a more detailed manipulation of encoding, maintenance and manipulation processes will be required to precisely pinpoint the functional contribution of the cerebellum in this domain. Nonetheless, the current findings already provide important constraints on the cerebellar role in working memory, demonstrating the utility of our selective recruitment approach for studying cerebellar function in cognition.

In terms of using this new approach to investigate cerebellar function, there are a number of important methodological factors to consider. First, the analysis heavily depends on the connectivity model that is used to predict the cerebellar activity. We addressed this issue by considering multiple models, including variations that allowed for more or less convergence (*King et al., 2023*). We also showed that the results hold when using a model that is trained on a larger number of datasets (*Nettekoven et al., 2024b*), a step that improved the overall predictive accuracy of the approach (see *Figure 3—figure supplement 1*). To be perfectly clear, we do believe that the task-invariant connectivity model is, as all models, ultimately *wrong* (*Box, 1976*). Indeed, our current study shows clear and systematic deviations from the model's prediction. Nonetheless, we consider it to be a *useful* model, in that it can serve as a strong null hypothesis, one that can be used to test for task-specific upregulation of activity. Therefore, our approach will benefit from further improvements of the model, such that it approximates the average cortico-cerebellar connectivity as closely and representatively as possible.

Second, the connectivity model, as it is currently constructed, does not predict the absolute level of cerebellar activity, but rather activity for one condition relative to other conditions. This limitation arises from the fact that the absolute magnitude of the BOLD signal in the cerebellum depends on many measurement-related factors, and the fact that we need to apply relatively heavy regularization to obtain good model performance. We therefore need to estimate the linear relationship between predicted and observed activity for each participant separately. Thus, our approach currently relies on the comparison to control conditions that activate similar neocortical regions to comparable extent, but recruit the cerebellum to a lesser degree.

Third, cortico-cerebellar connectivity is of course bidirectional. In our model, we do not model the influence of the cerebellum on neocortical activity, mediated through projections from the deep cerebellar nuclei to the thalamus. The simple reason for this decision is that cerebellar activity does not reflect the output firing of the Purkinje cells (*Caesar et al., 2003a*; *Thomsen et al., 2004*; *Thomsen et al., 2009*) and that, in contrast to cerebellar activity, cortical activity is determined by many other sources, including powerful cortico-cortical connectivity (*King et al., 2023*).

Finally, our approach does not allow us to conclude that the cerebellum is *not* necessary for a task when selective recruitment is *not* observed. Our approach simply shows that much of cerebellar activity can be fully accounted for by a task-invariant transmission of information from the neocortex, raising the possibility that this observed activity is an epiphenomenon of cortical input. Indeed, it would be very surprising if task-dependent gating was so complete that we would not see any activity in cerebellar circuits that receive input from activated cortical regions. Given this, we should expect some cerebellar activity even when the cerebellum makes minimal contributions to task performance (as observed in the force condition). Overall, we believe that showing task-specific violations on a task-invariant connectivity model provides much stronger evidence for a specific cerebellar role in a task than the mere presence of activity.

An important question for future study centers on elucidating the neurophysiological mechanisms that underlie task-dependent gating of cortical input to the cerebellum. One obvious candidate for gating are the pontine nuclei which integrate descending signals from different neocortical areas with feedback signals from the cerebellum (*Schwarz and Thier, 1999*). The cellular properties of pontine neurons are ideal for gating input signals in a state-dependent manner (*Möck et al., 1997*). Alternatively, gating could be achieved via modulation within the granule cell layer itself, perhaps via recurrent loops involving inhibitory Golgi cells (*Maex and De Schutter, 1998*). Violations of our connectivity model may also be caused by increased climbing fiber input under specific task conditions. Finally, gating may already occur in the neocortex: A recent study (*Park et al., 2022*) showed more recruitment of neocortical neurons that project to the pons when controlling the spatial aspects of joystick manipulation, and more recruitment of neurons that project intra-cortically or to the striatum when controlling movement amplitude. Because the neocortical BOLD signal reflects the activity of both neuronal populations, pontine-projecting neurons may be more engaged during fast alternating movements, even though the fMRI activity is the same as during the production of high forces.

Whichever combination of mechanisms is responsible for our observed effect, task-dependent gating of inputs to the cerebellum would be highly adaptive from a metabolic standpoint (*Attwell and Iadecola, 2002*): the costly mossy-fiber system would be most activated when cerebellar computation is required. For us as researchers, this gating phenomena offers a promising new keyhole that may allow us to unlock the use of fMRI for testing cerebellar contributions across cognitive tasks (*Diedrichsen et al., 2019*).

## Methods
### Participants
All participants gave informed consent under an experimental protocol approved by the Institutional Review Board at Western University (Protocol #107293). None of the participants reported a history of neurological or psychiatric disorders or current use of psychoactive medications. A total of 21 participants started the experiment. Of these, four participants were not scanned because of poor performance during the behavioral training session. The remaining 17 participants performed the tasks inside the scanner. The data for one participant were excluded due to an incidental finding. Therefore, the analyses were based on the data from 16 participants (8 females, 8 males, mean age = 25, std age = 2).

### Apparatus and stimuli
Participants used a custom-made 5-key finger keyboard to perform the finger tapping and digit span tasks. A force transducer, located under each key (FSG15N1A, Honeywell Sensing and Control; dynamic range, 0–25 N), continuously recorded the isometric force exerted by each finger at a rate of 500 Hz. We recalibrated each sensor (no force applied) at the beginning of each run to correct for drift. The applied force was continuously displayed to the participants in form of five short horizontal bars that moved along the vertical axis proportional to force exerted by each finger (*Figure 1A*: applied forces).

## Procedure

### Finger tapping task

Each trial was randomly selected from one of five conditions (*Table 1*). In all conditions, the response interval lasted for 6 s and participants were instructed to adopt a rate to distribute their responses evenly across this interval. For the Baseline condition, the target force was 2.5 N, and the instructed number of presses was 6 (i.e., optimal performance is 1 response/s). For the medium and high-force conditions, the target force was either 6 or 10 N, with the target number of presses fixed at 6. For the medium and high-speed conditions, the target number of presses was 10 or 18, with the target force fixed at 2.5 N.

A trial started with a short cueing phase (500 ms) during which two numeric characters (3 and 4) were presented on the screen, instructing the participant to tap with the right middle and ring finger. The required force level was indicated by a gray box that extended from 80% to 120% of the trial's target force (*Figure 1*, target force area), and the required number of presses by either 6, 10, or 18 small gray squares (*Figure 1*, instructed # taps).

After the 500 ms cueing phase, the two rectangles framing the digits turned from white to green, signaling to the participants to perform alternating finger presses. A horizontal green line (*Figure 1*, timer) started growing from left to right, indicating the passing of time. A press was registered when the force exceeded 80% of the target force (lower bound of the target force area). At this point, the force area changed color from gray to green and the color of the corresponding press square changed. When the force level returned to <1 N, the force area color changed back to gray.

After the response phase, participants received performance feedback. If the participant made the required number (±2) of alternating movements and completed the set of responses within 4–6 s, they received visual feedback indicating they had earned four points. This response time window was relatively liberal, because our main focus was not to match speeds exactly, but to get sufficient variation across conditions. All other outcomes were considered errors and were not rewarded (0 points). If the average exerted force for the trial exceeded 120% of the target force, the experimenter provided verbal feedback, asking the participant to press with less force. The message 'TOO FAST' was displayed if total movement time was shorter than 4 s or if the number of produced presses exceeded the instructed number by more than two. The message 'TOO SLOW' was displayed if the number of produced presses by the end of the 6-s interval was 3 or more below the instructed number of presses. Visual feedback (points or error message) remained on the screen for 500 ms. After a delay of 500 ms (inter-trial interval), the next trial began with the appearance of the next cue.

### Digit span task

Each trial started with a short cuing phase (500 ms), during which a red frame was presented outlining where the digits would appear along with a colored square on the left side that specified the recall direction (orange = forward recall; blue = backward recall). A white box within the red frame outlined the digits that would have to be remembered (2, 4, or 6). After the cue phase, a 6-s encoding phase started. Six digits were presented sequentially (1 s/digit) from left to right. The digits were drawn randomly (with replacement) from the set 1–5. The digits in the white box changed to a # symbol after 1 s; the other digits remained on the screen. For loads 2 and 4, the white box always encompassed the digits in the middle of the sequence (e.g., 13##45 or 1####5).

The encoding phase ended after the 1-s display time of the last digit and was followed by an additional 1-s delay. Following this, the procedure followed one of two paths. On No-Go trials, the screen blanked and 500 ms later, a new trial started with the cueing phase. These trials were included to be able to separate the activity associated with memory encoding and retrieval phases (see fMRI first-level analysis). On Go trials, the frame surrounding the digits turned green, indicating the start of the retrieval phase. Participants were instructed to press the key linked to the digit (1: thumb, 2: index, 3: middle, 4: ring, 5: pinky), either from memory or, for loads 2 and 4, from the visible digits on the screen. For forward trials (orange square), the participant was instructed to produce the responses to match the order observed in the encoding phase. For backwards trial (blue square), the participant was to reverse the sequential order of the digits, producing the right-most digit (last cued during encoding) first. For both conditions, the retrieval phase lasted for 7 s in total, giving participants enough time to complete the response (based on pilot work). To roughly match the speed of responding between the

very easy (forward load 2) and very difficult (backwards load 6) conditions, participants were instructed to evenly space their responses across the 7-s retrieval period.

Participants received visual feedback immediately after each response. If the response was correct, the corresponding hashtag or digit turned green, if incorrect, red. Only one response was allowed for each item. At the end of the retrieval phase, participants received additional feedback for 500ms summarizing trial performance (+4: all correct; +3: 1 error; +2: 2 errors; 0: otherwise). This point system was selected to encourage participants to attempt to recall each item.

### Experimental sessions

Each participant completed two sessions, a practice session conducted outside the scanner and a test session conducted in the scanner. Each session involved five runs of the finger tapping task interleaved with five runs of the digit span task. Each run of the finger tapping task consisted of 5 repetitions of each of the 5 conditions with the order randomized (total of 25 trials/run, approx. 5 min/run). Each run of the digit span task consisted of three Go trials and one No-Go trial for each of the six conditions (3 Set sizes × 2 Recall Directions) with the order fully randomize (total of 24 trials/run, approx. 8 min/run). The practice session was completed between 3 and 10 days prior to the scanning session.

## Image acquisition

MRI data were acquired on a 3T Siemens Prisma at the Center for Functional and Metabolic Mapping (CFMM) at Western University. A high-resolution whole-brain anatomical MPRAGE image was acquired at the beginning of the scanning session voxel size = 1 mm$^3$, field-of-view = 25.6 × 25.6 × 25.6 cm$^3$. Whole-brain functional images were acquired using an echo-planar imaging sequence with Repetition time (TR) = 1000 ms, Echo time (TE) = 30 ms, voxel size = 2.5 × 2.5 × 3 mm$^3$, field-of-view = 20.8 × 20.8 × 20.8 cm$^3$, 48 slices, P to A phase encoding direction, with multi-band acceleration factor = 3 (interleaved) and in-plane acceleration factor = 2. Gradient echo field maps were acquired to correct for distortions due to B0 inhomogeneities (acquisition parameters: voxel size = 3 × 3 × 3 mm$^3$, field-of-view = 24 × 24 × 24 cm$^3$). Physiological signals of heartbeat and respiration were recorded online during each functional run. Each functional run of the finger tapping task lasted ~5 min (260 volumes) and each run of the digit span task lasted for ~8 min (412 volumes).

## fMRI data processing

We used tools from SPM12 (*Friston et al., 1994*) and custom written code in MATLAB 2018b to process the functional and anatomical data. We defined an individual coordinate system for each subject by setting the origin of the anatomical image to the approximate location of the anterior commissure. Anatomical images were segmented into gray matter, white matter, csf, and skull. Functional images were corrected for head motion using the six-parameter rigid body transformation and were then co-registered to the individual anatomical image. The mean functional image and the results of anatomical segmentation were used to generate a gray matter mask for functional images. Slice timing correction, smoothing, and group normalization were not applied at this stage.

## fMRI first-level analysis

A first-level general linear model (GLM) was fit to the time series data of each run separately using SPM12. For the motor dataset, each condition was modeled as a separate regressor using a 6-s boxcar covering the response interval, convolved with a canonical hemodynamic response function (HRF). Error trials (approx. 5% of all trials) were modeled as one single regressor in the GLM and this regressor was discarded from further analysis.

For the working memory task, the encoding phase was modeled using a 7-s boxcar including 6 s of digit sequence display and the 1-s delay. The retrieval phase was modeled using a separate 7-s boxcar regressor covering the response interval. In Go trials, the two regressors therefore followed each other immediately, leading to a substantial correlation after the convolution with the HRF. However, the inclusion of 25% No-Go trials, for which only the encoding regressor was present, de-correlated the encoding and retrieval regressors sufficiently to enable stable and accurate estimate of the two processes. For analysis of imaging data, we chose to include all trials, including trials in which the participants made an error. We justify this given that there was no evidence indicating that any

participant ceased trying to do the task; as such, it is reasonable to assume that trials resulting in an error engaged the same memory processes.

Beta weights estimated by the first-level GLM were divided by residual-root-mean-square image, resulting in normalized activity estimates for each voxel, condition, and run. Rest was not modeled explicitly but served as an implicit baseline. Functional and anatomical data were transformed into a cortical and cerebellar atlas using a unified code framework (available on GitHub; copy archived at *Nettekoven et al., 2024a*).

## Cerebellar normalization

The cerebellum was isolated from the rest of the brain and segmented into white and gray matter using the Spatially Unbiased Infratentorial Template (SUIT) toolbox (*Diedrichsen, 2006*), followed in some cases by hand correction. Cerebellar white and gray matter probabilistic maps were deformed simultaneously into SUIT atlas space using a non-linear deformation algorithm (*Ashburner, 2007*). The deformation was applied to both anatomical images, and the normalized beta weights from the first-level GLM. Before normalization, the isolation mask was applied to discard the influence of adjacent inferior and occipital neocortical areas. For visualizations, the functional maps were projected onto a flat representation of the cerebellum (*Diedrichsen and Zotow, 2015*) using the SUIT toolbox.

## Neocortical normalization

For each participant, the anatomical image was used to reconstruct neocortical white matter and pial surface using Freesurfer (*Fischl, 2012*). Reconstructed surfaces were inflated to a sphere and registered to the fsLR 32 k node template (*Van Essen et al., 2012*) using a sulcal-depth map and local curvature. Neocortical activity patterns were projected onto these surfaces by averaging the activation values of voxels touching the line between corresponding vertices of the individual white matter and pial surface.

## ROI selection

For both datasets, we used a new symmetric functional atlas of the human cerebellum (*Nettekoven et al., 2024b*) that integrates data from seven large task-based datasets. The regions within this parcellation were estimated using a hierarchical Bayesian approach, with the constraint that the boundaries between regions were symmetric in the left and right hemispheres. For the motor dataset, we focused on right M3, a subregion of the motor domain that shows high selectivity for right-hand movements. For the working memory dataset, we focused on right D3, a subregion of the multi-demand network that that showed the clearest response to verbal digit span and verbal *N*-back tasks in the training data for the atlas.

## Connectivity model

We used task-invariant models of cortico-cerebellar connectivity to predict the activity pattern in the cerebellar ROI given the activity pattern in the cerebral cortex (*King et al., 2023*). This served as the null model from which we could evaluate the deviations in activity patterns, the test of the selective recruitment hypothesis. The models were trained on a large dataset with $N = 24$ subjects, each of whom was scanned for ~6 hr using two sets of tasks spanning a large range of motor and cognitive domains (MDTB; *King et al., 2019*). Each task set was performed in two sessions. For each participant, the neocortical surface was subdivided using regular icosahedron parcellations of different granularities, resulting in $P = 80$–1848 parcels. The normalized activity estimates (see first-level analysis) for all $N$ conditions were then averaged within each parcel and collected into a $N \times P$ matrix. These neocortical activations served as the predictors in the model ($\mathbf{X}$). The normalized activity estimates for the cerebellum were extracted in SUIT space at an isotropic resolution of 3 mm, resulting in an $N \times Q$ ($29 \times 6918$) matrix ($\mathbf{Y}$). We estimated the $P \times Q$ matrix of connectivity weight ($\mathbf{W}$) by minimizing the square error of the linear regression model $\mathbf{Y} = \mathbf{X} \mathbf{W} + \mathbf{E}$. To regularize this underspecified estimation problem, we employed either L1 regularization (Lasso) or L2-regualrizarion (Ridge regression). Hyperparameters were tuned using fivefold cross-validation within the training data (see *King et al., 2023* for details).

The models were trained on the first task set ($N = 29$ task conditions) and evaluated on the second task set of the MDTB ($N = 32$ different task conditions acquired from the same participants). Predictive

accuracy of the model was defined as the Pearson correlation between the observed and predicted response profile of each voxel across the tasks. For the present paper, we selected the model with the highest predictive accuracy across subjects, a ridge-regression model with a regularization parameter of $\lambda = exp$ (8) and 1848 neocortical parcels/predictors. We also used the Lasso model $\lambda = exp$ (−5), 1848 neocortical parcels, to assess the generality of the results.

Finally, we also tested an improved connectivity model that was obtained by integrating data from five task-based datasets (including MDTB), totaling 376 task conditions, 87 subjects, and 383 hr of imaging data (*Nettekoven et al., 2024b*). These new connectivity models were optimized and estimated on the individual subject level within each dataset, using L2-regularized regression (Ridge), and then averaged across all subjects and datasets. The data from the current study were not included in the derivation of these connectivity models.

To generate the predicted activity pattern, we used group-averaged connectivity weights for each voxel in the cerebellar ROI. We extracted the functional neocortical data by averaging the individual data within each of the 1848 neocortical parcel. As rest was not modeled explicitly in our first-level analyses, we added the resting baseline as a row of zeros to both the cortical and cerebellar data. In this way, the connectivity model was required to simultaneously predict the differences between conditions and the differences between each condition and rest. The matrix of individual neocortical activations was multiplied using the group-averaged connectivity weights to arrive at an individual prediction for each cerebellar voxel.

Given that connectivity weights were derived from a different dataset with different subjects and different signal-to-noise ratios (SNR), we fitted a simple linear regression line for each participant between the observed cerebellar activation and the model predictions for the selected ROI. The slope of the line accounts for differences in SNR between the two datasets. Even though rest is not shown in *Figures 3B and 6B*, it was included as a datapoint at (0,0) for the regression analysis. The residuals from this regression analysis were used for statistical testing across participants.

## Additional information

### Competing interests

Richard B Ivry: co-founder with equity in Magnetic Tides, Inc. Jörn Diedrichsen: Reviewing editor, *eLife*. The other authors declare that no competing interests exist.

### Funding

| Funder | Grant reference number | Author |
|---|---|---|
| Canadian Institutes of Health Research | PJT 159520 | Jörn Diedrichsen |
| Canadian Institutes of Health Research | PJT-191815 | Jörn Diedrichsen |
| Canada First Research Excellence Fund | BrainsCAN | Jörn Diedrichsen |
| Raynor Cerebellum Project | | Jörn Diedrichsen |
| National Institutes of Health | NS116883 | Richard B Ivry |
| National Institutes of Health | NS105839 | Richard B Ivry |

The funders had no role in study design, data collection, and interpretation, or the decision to submit the work for publication.

### Author contributions

Ladan Shahshahani, Conceptualization, Resources, Data curation, Software, Formal analysis, Validation, Investigation, Visualization, Methodology, Writing - original draft, Project administration, Writing – review and editing; Maedbh King, Resources, Software, Writing – review and editing; Caroline Nettekoven, Resources, Writing – review and editing; Richard B Ivry, Conceptualization, Writing – review

and editing; Jörn Diedrichsen, Conceptualization, Resources, Data curation, Software, Formal analysis, Supervision, Funding acquisition, Validation, Investigation, Visualization, Methodology, Writing - original draft, Project administration, Writing – review and editing

### Author ORCIDs
Ladan Shahshahani (iD) http://orcid.org/0000-0001-6189-9994
Maedbh King (iD) http://orcid.org/0000-0001-5374-1011
Richard B Ivry (iD) https://orcid.org/0000-0003-4728-5130
Jörn Diedrichsen (iD) http://orcid.org/0000-0003-0264-8532

### Ethics
All participants gave informed consent under an experimental protocol approved by the Institutional Review Board at Western University (Protocol #107293).

Reviewer #1 (Public Review): https://doi.org/10.7554/eLife.96386.3.sa1
Reviewer #2 (Public Review): https://doi.org/10.7554/eLife.96386.3.sa2
Author response https://doi.org/10.7554/eLife.96386.3.sa3

## Additional files

### Supplementary files
• MDAR checklist

### Data availability
The raw behavioral and imaging are available on OpenNeuro. The code for data management can be found here (copy archived at *Nettekoven et al., 2024a*). The connectivity models used in this paper and instructions of how to generate predictions for new data are available here (copy archived at *Diedrichsen et al., 2024*). The code for the analyses presented in the current paper is available here (copy archived at *Shahshahani et al., 2024*).

The following dataset was generated:

| Author(s) | Year | Dataset title | Dataset URL | Database and Identifier |
|---|---|---|---|---|
| Shahshahani L, King MB, Nettekoven C, Ivry RB, Diedrichsen J | 2024 | WMFS | https://doi.org/10.18112/openneuro.ds005148.v1.1.0 | OpenNeuro, 10.18112/openneuro.ds005148.v1.1.0 |

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
