## [Editor Report · eLife assessment]

This **important** study reports a novel approach to studying cerebellar function based on the idea of selective recruitment using fMRI. It provides **convincing** evidence for task-dependent gating of neocortical input to the cerebellum during a motor task and a working memory task. The study will be of interest to a broad cognitive neuroscience audience.

---

## [Referee Report · Reviewer #1 (Public Review)]

This is an interesting and well-written paper reporting on a novel approach to studying cerebellar function based on the idea of selective recruitment using fMRI. The study is well-designed and executed. Analyses are sound and results are properly discussed. The paper makes a significant contribution to broadening our understanding of the role of cerebellum in human behavior.

In the revision, the authors did an excellent job in addressing my concerns.

---

## [Referee Report · Reviewer #2 (Public Review)]

Summary:

Shahshahani and colleagues used a combination of statistical modelling and whole-brain fMRI data in an attempt to separate the contributions of cortical and cerebellar regions in different cognitive contexts.

Strengths:

* The manuscript uses a sophisticated integration of statistical methods, cognitive neuroscience and systems neurobiology.

* The authors use multiple statistical approaches to ensure robustness in their conclusions.

* The consideration of the cerebellum as not a purely 'motor' structure is excellent and important.

Weaknesses:

* The assumption that cortical BOLD responses in cognitive tasks should be matched irrespective of cerebellar involvement does not cohere directly with the notion of 'forcing functions' introduced by Houk and Wise, suggesting the need for future work.

---

## [Author Response]

The following is the authors’ response to the original reviews.

**Reviewer #1 (Public Review):**
This is an interesting and well-written paper reporting on a novel approach to studying cerebellar function based on the idea of selective recruitment using fMRI. The study is well-designed and executed. Analyses are sound and results are properly discussed. The paper makes a significant contribution to broadening our understanding of the role of the cerebellum in human behavior.

We thank the reviewer for the positive assessment of our paper.

(1) While the authors provide a compelling case for the link between BOLD and the cerebellar cortical input layer, there remains considerable unexplained variance. Perhaps the authors could elaborate a bit more on the assumption that BOLD signals mainly reflect the input side of the cerebellum (see for example King et al., elife. 2023 Apr 21;12:e81511).

Our paper is based on the assumption that the cerebellar BOLD signal reflects solely the input to the cerebellum and does not reflect the changes in firing rates of Purkinje cells. This assumption relies on two lines of arguments: Studies that have directly looked at the mechanism of vasodilation in the cerebellum, and studies that try to infer the contributions of different neurophysiological mechanisms to overall cerebellar metabolism (Attwell and Iadecola, 2002).

Vasodilatory considerations: The mechanisms that causes vasodilation in the cerebellum, and hence BOLD signal increases, has been extensively studied: Electrical stimulation of mossy fibers (Gagliano et al., 2022; Mapelli et al., 2017), as well as parallel fibers (Akgören et al., 1994; Iadecola et al., 1996; Mathiesen et al., 1998; Yang and Iadecola, 1997) lead to robust increases in cerebellar blood flow. In contrast to the neocortex, the regulation of blood flow in the cerebellum depends nearly purely on the vasodilator Nitric Oxide (NO) (Akgören et al., 1994; Yang and Iadecola, 1997) with stellate cells playing a key role in the signaling cascade (Yang et al., 2000).

Electrical (Mathiesen et al., 2000) and pharmacological (Yang and Iadecola, 1998) stimulation of climbing fibers also leads to robust increases in blood flow. Simultaneous parallel and climbing fiber stimulation seems to combine sub-additively to determine the blood flow changes (10).

Importantly, even dramatic changes in spiking rate of Purkinje cells do not lead to changes in vasodilation. For starters, parallel fiber stimulation leads to blood flow increases, even though the net effect on Purkinje cell firing is inhibitory (Mathiesen et al., 1998). More importantly, complete inhibition of the Purkinje cell using a GABA agonist does not change baseline cerebellar blood flow (10). Conversely, even a 200-300% increase in simple (and complex) spike firing rate through application of a GABA antagonist does not show any measurable consequences for blood flow, even though it clearly increases the metabolic rate of oxygen consumption in the tissue (Thomsen et al., 2009, 2004).

In sum, this extensive set of studies clearly argues that the cerebellar blood flow response is mostly dictated by synaptic input, and that the firing rate of Purkinje cells does not influence vasodilation. Because the BOLD signal is caused by an supply of oxygen over and above the level of oxygen consumption, this would argue that increases in Purkinje cell firing would not lead to BOLD increases. What is less clear is the degree to which changes in BOLD signal during normal activity are determined by changes in mossy fiber or climbing fiber input. Disruption of either pathway leads to 60-70% reductions in the evoked blood flow response during whisker stimulation (Yang et al., 2000; Zhang et al., 2003) – but it remains unclear to what degree this reflects the distribution of contributions in the healthy animal, as these powerful disruptions may have a number of side-effects.

Metabolic considerations: To estimate the relative contributions climbing fiber / mossy fiber input to the variations in BOLD signal under natural conditions, it is useful to consider the contributions of different cerebellar processes to the overall metabolism of the cerebellum. Assuming an average firing rate of 40Hz for mossy fibers, ~3Hz for Granule cells, and 1Hz for climbing fibers, Howarth et al. (Howarth et al., 2012, 2010) estimated that the transmission from mossy fibers to granular cells, dominates the energy budget with 53%. The subsequent stage, encompassing the transfer of information from Granular cells to Purkinje cells, accounts for 32% of energy expenditure. In contrast, integration within Purkinje cells and the spiking (simple and complex) of these cells represents only 15% of the total energy consumption.

More important for the BOLD signal, however, are the activity-induced variations in metabolic consumption: Purkinje cells fire relatively constantly at a very high frequency (~50Hz) both during awake periods and during sleep (Shin et al., 2007). When providing a signal to the neocortex, firing rate decreases, actually lowering the metabolic demand. Climbing fibers normally fire at ~0.5 Hz and even during activity rarely fire much above 2Hz (Streng et al., 2017). In contrast, granule cells show a low firing rates during rest (typically <1hz) and can spike during activity well above 100Hz. Combined with the sheer number of granule cells, these considerations would suggest that the vast majority of the variation in metabolic demand are due to mossy fiber input and granule cell activity.

Overall, we therefore think it is likely that the main determinant of the cerebellar cortical BOLD signal is mossy fiber input and the transmission of information from mossy fibers to granule cells to Purkinje cells. We admit that the degree to which climbing fiber input contribute to BOLD signal changes is much less clear. We can be quite certain, however, that the firing rate of Purkinje cells does not contribute to the cerebellar BOLD signal, as even dramatic changes in the firing rate do not cause any changes in vasodilation. We have clarified our line of reasoning in the paper, and hope this more extensive response here will give the reader a better overview over the pertaining literature.

(2) The current approach does not appear to take the non-linear relationships between BOLD and neural activity into account.

Thank you for raising this concern. We did not stress this point in the paper, but one big advantage of our selective recruitment approach is that it is – to some degree- robust against non-linearities in the relationship between neural activity and BOLD signal. This is the case, as long as the shape of the non-linearity is similar in the cerebellum and the neocortex. The results of our motor task (Figure 3) provide a clear example of this: The BOLD signal both in the neocortex and cerebellum incases non-linearly as a function of force – the increase from 2.5N to 6N (a 3.5N increase) is larger than the increase from 6N to 10N (a 4N increase). A similar non-linearity can be observed for tapping speed (6, 10 to 18 taps / s). However, within each condition, the relationship between cortical and cerebellar activity is nearly perfectly linear, reflecting the fact that the shape of the non-linearity for the cerebellum and cortex is very similar.

Most importantly, even if the non-linearity across the two structures is different, any non-linear relationship between neural activity and BOLD signal (of vasodilatory nature) should apply to different conditions (here force and speed increases) similarly. Therefore, if two conditions show overlapping activity levels (as observed for force and speed across medium and high levels, Figure 3), a offset between conditions cannot be caused by a non-linearity in the relationship of cortical and cerebellar activity. Because all conditions are subject to the same non-linearity, all points should lie on a single (likely monotonically increasing) non-linear function. Both for the motor and working memory task, the pattern of results clearly violates this assumption.

(3) The authors may want to address a bit more the issue of closed loops as well as the underlying neuroanatomy including the deep cerebellar nuclei and pontine nuclei in the context of their current cerebello-cortical correlational approach. But also the contribution of other brain areas such as the basal ganglia and hippocampus.

Cortical-cerebellar communication is of course bi-directional. As discussed in King at al., (2023), however, we are restricting our model to the connections from the neocortex to the cerebellum for the following reasons: First, cerebellar BOLD activity likely reflects mostly neocortical input (see our answer to pt. 1), whereas neocortical activity is determined by a much wider array of projections, including striato-thalamo-cortical and cortico-cortical connections. Secondly, the output of the cerebellum cannot be predicted from the BOLD signal of the cerebellar cortex, as it is unlikely that the firing rate of Purkinje cells contribute to cerebellar BOLD signal (see pt. 1). For these reasons we believe that the relationship between neocortical and cerebellar activity patterns is mostly dictated by the connectivity from cortex to cerebellum, and is therefore best modelled as thus. This is now more clearly discussed in a new paragraph (line 318-323) of the revised manuscript.

We are also ignoring other inputs to the cerebellum, including the spinal chord, the basal ganglia (Bhuvanasundaram et al., 2022; Bostan and Strick, 2018) hippocampus (Froula et al., 2023; Watson et al., 2019), and amygdala (Farley et al., 2016; Jung et al., 2022; Terburg et al., 2024). In humans, however, the neocortex remains the primary source of input to pontine nuclei. Consequently, it stands as the main structure shaping activity within the cerebellar cortex. While it is an interesting question to what degree the consideration of subcortical structures can improve the prediction of cerebellar activity patterns, we believe that considering the neocortex provides a good first approximation.

**Reviewer #1 (Recommendations):**
(4) A few sentences to clarify the used models as was done in the King et al. (2024) paper may improve readability.

We have now added the sentences in the introduction (line 25ff):

To approach this problem, we have recently developed and tested a range of cortical-cerebellar connectivity models (King et al., 2023), designed to capture fixed, or task-invariant, transmission between neocortex and cerebellum. For each cerebellar voxel, we estimated a regularized multiple regression model to predict its activity level across a range of task conditions (King et al., 2019) from the activity pattern observed in the neocortex for the same conditions. The models were then evaluated in their ability to predict cerebellar activity in novel tasks, again based only on the corresponding neocortical activity pattern. Two key results emerged from this work. First, while rs-FC studies (Buckner et al., 2011; Ji et al., 2019; Marek et al., 2018) have assumed a 1:1 mapping between neocortical and cerebellar networks, models which allowed for convergent input from multiple neocortical regions to a single cerebellar region performed better in predicting cerebellar activity patterns for novel tasks. Second, when given a cortical activation pattern, the best performing model could predict about 50% of the reliable variance in the cerebellar cortex across tasks (King et al., 2023).

(5) To what extent does this paper demonstrate the limitations of BOLD in neuroscientific research?

The primary objective of this study was to shed light on the problems of interpreting BOLD activation within the cerebellum. The problem that the BOLD signal mostly reflect input to a region is not unique to the cerebellum, but also applies (albeit likely to a lesser degree) to other brain structures. However, the solution we propose here critically hinges on three features of the cerebellar circuitry: (a) the mossy fiber input for the cerebellar hemispheres mostly arise from the neocortex, (b) the BOLD signal is likely dominated by this mossy fiber input (see pt. 1), and (c) there is very little excitatory recurrent activity in the cerebellum, so output activity in the cerebellum does not cause direct activity in other parts of the cerebellum.

These features motivate us to use a directed cortex->cerebellum connectivity model, which does not allow for any direct connectivity within the cerebellum. While the same approach can also be applied to other brain structures, it is less clear that the approach would yield valid results here. For example, due the local excitatory recurrent connectivity within neocortical columns, the activity here will also relate to local processing.

(6) What if the authors reversed their line of reasoning as in that cerebellum activity is matched to map changes in cerebral cortical activity? Perhaps this could provide further evidence for the assumed directional specificity of the task-dependent gating of neocortical inputs.

Given (a) that the cerebellar BOLD signal tells us very little about cerebellar output signals (b) that there are many other input signals to the neocortex that are more powerful than cerebellar inputs, and (c) that there strong cortical-cortical connections, we believe that this model would be hard to interpret (see also our answer to pt. 3).

Therefore, while the inversion of the linear task-invariant mapping between cortical and cerebellar activity is a potentially interesting exercise, it is unclear to us at this point what strong predictions we would be able to test with this approach.

(7) The statement that cerebellar fMRI activity may simply reflect the transmission of neocortical activity through fixed connections can be better explained. Also in the context of using the epiphenomenon (on page 11) in the paper. To what extent is the issue of epiphenomenon not a general problem of fMRI research?

We have rephrased the introduction of this idea (line 17):

This means that increases in the cerebellar BOLD signal could simply reflect the automatic transmission of neocortical activity through fixed anatomical connections. As such, whenever a task activates a neocortical region, the corresponding cerebellar region would also be activated, regardless of whether the cerebellum is directly involved in the task or not.

Epiphemonal activity: This is indeed a general problem in fMRI research (and indeed research that uses neurophysiological recordings, rather than manipulations of activity). Indeed, we have discussed similar issues in the context of motor activity in ipsilateral motor cortex (Diedrichsen et al., 2009). However, given that we only offer a possible approach to address this issue for the cerebellum (see pt. 5), we thought it best to keep the scope of the discussion focused on this structure.

**Reviewer #2 (Public Review):**
Summary:Shahshahani and colleagues used a combination of statistical modelling and whole-brain fMRI data in an attempt to separate the contributions of cortical and cerebellar regions in different cognitive contexts.Strengths:The manuscript uses a sophisticated integration of statistical methods, cognitive neuroscience, and systems neurobiology.The authors use multiple statistical approaches to ensure robustness in their conclusions.The consideration of the cerebellum as not a purely 'motor' structure is excellent and important.

We thank the reviewer for their positive evaluation.

Weaknesses:(1) Two of the foundation assumptions of the model - that cerebellar BOLD signals reflect granule cells > purkinje neurons and that corticocerebellar connections are relatively invariant - are still open topics of investigation. It might be helpful for the reader if these ideas could be presented in a more nuanced light.

Please see response to the comment 1 of Reviewer 1 for a more extensive and detailed justification of this assumption. We have now also clarified our rationale for this assumption better in the paper on line 10-14. Finally, we now also raise explicitly the possibility that some of the violations of the task-invariant model could be caused by selectively increase of climbing fiber activity in some tasks (line 340).

(2) The assumption that cortical BOLD responses in cognitive tasks should be matched irrespective of cerebellar involvement does not cohere with the idea of 'forcing functions' introduced by Houk and Wise.

We are assuming that you refer to the idea that cerebellar output is an important determinant of the dynamics (and likely also of the magnitude) of neocortical activity. We agree most certainly here. However, we also believe that in the context of our paper, it is justified to restrict the model to the connectivity between the neocortex and the cerebellum only (see reviewer 1, comment 3).

Furthermore, if increased cerebellar output indeed occurs during the conditions for which we identified unusually high cerebellar activity, it should increase neocortical activity, and bring the relationship of the cerebellar and cortical activity again closer to the predictions of the linear model. Therefore, the identification of functions for which cerebellar regions show selective recruitment is rather conservative.

**Reviewer #2 (Recommendations):**
(3) One of the assumptions stated in the abstract -- that the inputs to the cerebellum may simply be a somewhat passive relay of the outputs of the cerebral cortex -- has been challenged recently by work from Litwin-Kumar (Muscinelli et al., 2023 Nature Neuroscience), which argues for complex computational relationships between cortical pyramidal neurons, pontine nuclei and granule cells, which in turn would have a non-linear impact on the relationship between cortical and cerebellar BOLD. The modelling is based on empirical recordings from Wagner (2019, Cell) which show that the synaptic connections between the cortex and granule cells change as a function of learning, further raising concerns about the assumption that the signals inherent within these two systems should be identical. Whether these micro-scale features are indicative of the macroscopic patterns observed in BOLD is an interesting question for future research, but I worry that the assumption of direct similarity is perhaps not reflective of the current literature. The authors do speak to these cells in their discussion, but I believe that they could also help to refine the authors' hypotheses in the manuscript writ large.

We absolutely agree with your point. However, we want to make extremely clear here that our hypothesis (that the inputs to the cerebellum are a linear task-invariant function of the outputs of the cerebral cortex) is the Null-hypothesis that we are testing in our paper. In fact, our results show the first empirical evidence that task-dependent gating may indeed occur. In this sense, our paper is consistent with the theoretical suggestion of (Muscinelli et al., 2023).

You may ask whether a linear task-invariant model of cortical-cerebellar connectivity is not a strawman, given that is most likely incorrect. However, as we stress in the discussion (line 298-), a good Null-model is a useful model, even if it is (as all models) ultimately incorrect. Without it, we would not be able to determine which cerebellar activity outstrips the linear prediction. The fact that this Null-model itself can predict nearly 50% of the variance in cerebellar activity patterns across tasks at a group level, means that it is actually a very powerful model, and hence is a much more stringent criterion for evidence for functional involvement than just the presence of activity.

(4) Further to this point, I didn't follow the authors' logic that the majority of the BOLD response in the cerebellum is reflective of granule cells rather than Purkinje cells. I read through each of the papers that were cited in defense of the comment: "The cerebellar BOLD signal is dominated by mossy fiber input with very little contribution from the output of the cerebellar cortex, the activity of Purkinje cells" and found that none of these studies made this same direct conclusion. As such, I suggest that the authors soften this statement, or provide a different set of references that directly confirm this hypothesis.

Please see response to the comment 1, Reviewer 1. We hope the answer provides a more comprehensive overview over the literature, which DOES show that spiking behavior of Purkinje cells does not influence vasodilation (as opposed to mossy fiber input). We have now clarified our rationale and the exact cited literature on line 9-14 of the paper.

(5) Regarding the statement: "As such, whenever a task activates a neocortical region, we might observe activity in the corresponding cerebellar regions regardless of whether the cerebellum is directly involved in the task or not." -- what if this is a feature, rather than a bug? That is, the organisation of the nervous system has been shaped over phylogeny such that every action, via efference copies of motor outputs, is filtered through the complex architecture of the cerebellum in order to provide a feed-forward signal to the thalamus/cortex (and other connected structures). Houk and Wise made compelling arguments in their 1995 Cerebral Cortex paper arguing that these outputs (among other systems) could act as 'forcing functions' on the kinds of dynamics that arise in the cerebral cortex. I am inclined to agree with their hypothesis, where the implication is that there are no tasks that don't (in some way) depend on cerebellar activity, albeit to a lesser or greater extent, depending on the contexts/requirements of the task. I realise that this is a somewhat philosophical point, but I do think it is important to be clear about the assumptions that form the basis of the reasoning in the paper.

This is an interesting point. Our way of thinking about cerebellar function does indeed correspond quite well to the idea of forcing functions- the idea that cerebellar output can “steer” cortical dynamics in a particular way. However, based on patient and lesion data, it is also clear that some cortical functions rely much more critically on cerebellar input than others. We hypothesize here that cerebellar activity is higher (as compared to the neocortical activity) when the functions require cerebellar computation.

We also agree with the notion that cerebellar contribution is likely not an all-or-none issue, but rather a matter of gradation (line 324ff).

(6) Regarding the logic of expecting the cortical patterns for speed vs. force to be matched -- surely if the cerebellum was involved more in speed than force production, the feedback from the cerebellum to the cortex (via thalamus) could also contribute to the observed differences? How could the authors control for this possibility?

Our model currently indeed does not attempt to quantify the contributions of cerebellar output to cortical activity. However, given that cerebellar output is not visible in the BOLD signal of the cerebellum (see reviewer 1, comment 1), we believe that this is a rational approach. As argued in our response to your comment 2, increased cerebellar output in the speed compared to the force condition should bring the activity relationship closer to the linear model prediction. The fact that we find increased cerebellar (as compared to neocortical) activity in the speed conditions, suggests that there is indeed task-dependent gating of cortical projections to the cerebellum.

Akgören N, Fabricius M, Lauritzen M. 1994. Importance of nitric oxide for local increases of blood flow in rat cerebellar cortex during electrical stimulation. *Proc Natl Acad Sci U S A* 91:5903–5907.

Attwell D, Iadecola C. 2002. The neural basis of functional brain imaging signals. *Trends Neurosci* 25:621–625.

Bhuvanasundaram R, Krzyspiak J, Khodakhah K. 2022. Subthalamic Nucleus Modulation of the Pontine Nuclei and Its Targeting of the Cerebellar Cortex. *J Neurosci* 42:5538–5551.

Bostan AC, Strick PL. 2018. The basal ganglia and the cerebellum: nodes in an integrated network. *Nat Rev Neurosci* 19:338–350.

Buckner RL, Krienen FM, Castellanos A, Diaz JC, Yeo BTT. 2011. The organization of the human cerebellum estimated by intrinsic functional connectivity. *J Neurophysiol* 106:2322–2345.

Caesar K., Gold L, Lauritzen M. 2003. Context sensitivity of activity-dependent increases in cerebral blood flow. *Proc Natl Acad Sci U S A* 100:4239–4244.

Caesar K., Thomsen K, Lauritzen M. 2003. Dissociation of spikes, synaptic activity, and activity-dependent increments in rat cerebellar blood flow by tonic synaptic inhibition. *Proc Natl Acad Sci U S A* 100:16000–16005.

Farley SJ, Radley JJ, Freeman JH. 2016. Amygdala Modulation of Cerebellar Learning. *J Neurosci* 36:2190–2201.

Froula JM, Hastings SD, Krook-Magnuson E. 2023. The little brain and the seahorse: Cerebellar-hippocampal interactions. *Front Syst Neurosci* 17:1158492.

Gagliano G, Monteverdi A, Casali S, Laforenza U, Gandini Wheeler-Kingshott CAM, D’Angelo E, Mapelli L. 2022. Non-linear frequency dependence of neurovascular coupling in the cerebellar cortex implies vasodilation-vasoconstriction competition. *Cells* 11:1047.

Howarth C, Gleeson P, Attwell D. 2012. Updated energy budgets for neural computation in the neocortex and cerebellum. *J Cereb Blood Flow Metab* 32:1222–1232.

Howarth C, Peppiatt-Wildman CM, Attwell D. 2010. The energy use associated with neural computation in the cerebellum. *J Cereb Blood Flow Metab* 30:403–414.

Iadecola C, Li J, Xu S, Yang G. 1996. Neural mechanisms of blood flow regulation during synaptic activity in cerebellar cortex. *J Neurophysiol* 75:940–950.

Ji JL, Spronk M, Kulkarni K, Repovš G, Anticevic A, Cole MW. 2019. Mapping the human brain’s cortical-subcortical functional network organization. *Neuroimage* 185:35–57.

Jung SJ, Vlasov K, D’Ambra AF, Parigi A, Baya M, Frez EP, Villalobos J, Fernandez-Frentzel M, Anguiano M, Ideguchi Y, Antzoulatos EG, Fioravante D. 2022. Novel Cerebello-Amygdala Connections Provide Missing Link Between Cerebellum and Limbic System. *Front Syst Neurosci* 16:879634.

King M, Hernandez-Castillo CR, Poldrack RA, Ivry RB, Diedrichsen J. 2019. Functional boundaries in the human cerebellum revealed by a multi-domain task battery. *Nat Neurosci* 22:1371–1378.

King M, Shahshahani L, Ivry RB, Diedrichsen J. 2023. A task-general connectivity model reveals variation in convergence of cortical inputs to functional regions of the cerebellum. *Elife* 12:e81511.

Mapelli L, Gagliano G, Soda T, Laforenza U, Moccia F, D’Angelo EU. 2017. Granular layer neurons control cerebellar neurovascular coupling through an NMDA receptor/NO-dependent system. *J Neurosci* 37:1340–1351.

Marek S, Siegel JS, Gordon EM, Raut RV, Gratton C, Newbold DJ, Ortega M, Laumann TO, Adeyemo B, Miller DB, Zheng A, Lopez KC, Berg JJ, Coalson RS, Nguyen AL, Dierker D, Van AN, Hoyt CR, McDermott KB, Norris SA, Shimony JS, Snyder AZ, Nelson SM, Barch DM, Schlaggar BL, Raichle ME, Petersen SE, Greene DJ, Dosenbach NUF. 2018. Spatial and Temporal Organization of the Individual Human Cerebellum. *Neuron* 100:977-993.e7.

Mathiesen C, Caesar K, Akgören N, Lauritzen M. 1998. Modification of activity-dependent increases of cerebral blood flow by excitatory synaptic activity and spikes in rat cerebellar cortex. *J Physiol* 512 ( Pt 2):555–566.

Mathiesen C, Caesar K, Lauritzen M. 2000. Temporal coupling between neuronal activity and blood flow in rat cerebellar cortex as indicated by field potential analysis. *J Physiol* 523:235–246.

Muscinelli SP, Wagner MJ, Litwin-Kumar A. 2023. Optimal routing to cerebellum-like structures. *Nat Neurosci* 26:1630–1641.

Shin S-L, Hoebeek FE, Schonewille M, De Zeeuw CI, Aertsen A, De Schutter E. 2007. Regular patterns in cerebellar Purkinje cell simple spike trains. *PLoS One* 2:e485.

Streng ML, Popa LS, Ebner TJ. 2017. Climbing Fibers Control Purkinje Cell Representations of Behavior. *J Neurosci* 37:1997.

Terburg D, van Honk J, Schutter DJLG. 2024. Doubling down on dual systems: A cerebellum–amygdala route towards action- and outcome-based social and affective behavior. *Cortex* 173:175–186.

Thomsen K, Offenhauser N, Lauritzen M. 2004. Principal neuron spiking: neither necessary nor sufficient for cerebral blood flow in rat cerebellum. *J Physiol* 560:181–189.

Thomsen K, Piilgaard H, Gjedde A, Bonvento G, Lauritzen M. 2009. Principal cell spiking, postsynaptic excitation, and oxygen consumption in the rat cerebellar cortex. *J Neurophysiol* 102:1503–1512.

Watson TC, Obiang P, Torres-Herraez A, Watilliaux A, Coulon P, Rochefort C, Rondi-Reig L. 2019. Anatomical and physiological foundations of cerebello-hippocampal interaction. *Elife* 8:e41896.

Yang G, Huard JM, Beitz AJ, Ross ME, Iadecola C. 2000. Stellate neurons mediate functional hyperemia in the cerebellar molecular layer. *J Neurosci* 20:6968–6973.

Yang G, Iadecola C. 1998. Activation of cerebellar climbing fibers increases cerebellar blood flow: role of glutamate receptors, nitric oxide, and cGMP. *Stroke* 29:499–507; discussion 507-8.

Yang G, Iadecola C. 1997. Obligatory role of NO in glutamate-dependent hyperemia evoked from cerebellar parallel fibers. *Am J Physiol* 272:R1155-61.

Zhang Y, Forster C, Milner TA, Iadecola C. 2003. Attenuation of activity-induced increases in cerebellar blood flow by lesion of the inferior olive. *Am J Physiol Heart Circ Physiol* 285:H1177-82.